# Haplotype Diversity of NADPH-Cytochrome P450 Reductase Gene of *Ophiocordyceps sinensis* and the Effect on Fungal Infection in Host Insects

**DOI:** 10.3390/microorganisms8070968

**Published:** 2020-06-29

**Authors:** Zixian Xu, Yunguo Zhu, Lingyan Xuan, Shan Li, Zhou Cheng

**Affiliations:** School of Life Science and Technology, Tongji University, Shanghai 200092, China; xuzixian1995@outlook.com (Z.X.); ygzhu@tongji.edu.cn (Y.Z.); xly1731505@163.com (L.X.)

**Keywords:** *Ophiocordyceps sinensis*, NADPH-cytochrome P450 reductase, gene haplotype, infection, host insects

## Abstract

*Ophiocordyceps sinensis* Berk. is a fungal parasite that parasitizes the larvae of Hepialidae and is used as a traditional Chinese medicine. However, it is not clear how *O. sinensis* infects its host. The encoding gene haplotype diversity and predicted function of the nicotinamide adenine dinucleotide phosphate (NADPH)-cytochrome P450 reductase (CPR) related to the fungal pathogenicity was analyzed for 219 individuals from 47 *O. sinensis* populations. Two NADPH CPR genes of *O. sinensis* were detected and their dominant haplotypes were widely distributed throughout the entire distribution range in Western China. Only 5.43% of all *O. sinensis* individuals possessed the specific private haplotypes of NADPH CPR-1 and CPR-2 genes. Bioinformatic analyses predicted that the phosphorylation sites, motifs, and domains of NADPH CPR of *O. sinensis* were different between those encoding by the dominant and private gene haplotypes. The one-to-one match fungus–host correspondence of the same individual suggested that the widely distributed *O. sinensis* with the dominant NADPH CPR gene haplotypes may strongly infect almost all host insects through a random infection by oral or respiratory pores. Conversely, *O. sinensis* with the specific private NADPH CPR gene haplotypes is likely to infect only a few corresponding host insects by breaching the cuticle, due to the changed NADPH CPR structure and function.

## 1. Introduction

*Ophiocordyceps sinensis* Berk., a fungal parasite which parasitizes the larvae of Hepialidae, forms a complex containing fungal sexual stroma and the caterpillar body [1,2]. *O. sinensis* has been used for centuries as a traditional Chinese medicine to treat asthma, bronchial/lung infection, and kidney disease [1]. Generally, most entomogenous fungi infect the insects through the cuticle, while some encroach the host orally and make their way through the gut [3]. As entomogenous fungi, *O. sinensis* may infect the host insect specifically through the cuticle or randomly through the oral cavity. *O. sinensis* produces ascospores, which become filamentous mycelium and intrude into the haemolymph of insects through the gender cuticle [4]. After infection, the filamentous mycelium fill the coeloms of the host larvae, resulting in the destruction of the haemolymph and death of the insects [4,5]. A study by nested PCR has indicated that *O. sinensis* appears in plant roots during its anamorph life cycle, presenting an increased opportunity to infect the larvae through the oral cavity [6]. A stable carbon isotope analysis has shown that the fungus growth is closely related to the digestive tract of its host larva, which also suggests that phytophagous larvae may become infected as they feed [7]. *O. sinensis* may have different infection strategies in different seasons. The main infection strategy is through the cuticle in the winter, when the host larvae are in dormancy [8]. In the summer, when the host insect sheds its cuticles [9], the host larvae are prone to the infection by *O. sinensis* from the habitat soils [10,11]. However, the mode of infection of *O. sinensis* into the host insects remains unclear.

At present, two independent studies on the whole-genome sequencing of *O. sinensis* have been completed, with opposing results about fungal pathogenicity [3,12]. In one study, *O. sinensis* displays a considerable lineage-specific expansion of gene families, such as cytochrome P450, which has functionally enriched in the adaptability of fungal pathogenicity and specialized host infection [12]. Meanwhile, another study indicates that *O. sinensis* gene families encoding the cytochrome P450 (CYP) subfamily CYP52 enzymes, cuticle-degrading proteases and chitinases, are greatly reduced in size. Additionally, protein families involved in adhesion to cuticles and the formation of appressoria are absent or reduced in *O. sinensis*. These results suggest that *O. sinensis* no longer breaches the intact cuticle and, instead, probably infects insects orally or via the spiracles [3].

Cytochrome P450 forms a small electron transfer chain, together with the nicotinamide adenine dinucleotide phosphate (NADPH) cytochrome P450 reductase (CPR), which is a flavoprotein containing flavin cofactors flavin adenine dinucleotide (FAD) and flavin mononucleotide (FMN) [13,14]. They can catalyze most oxidative reactions and have been shown to be involved in fungal pathogenicity [12]. The P450 mono-oxygenase encoding gene *bcbot1* is involved in the botrydial biosynthesis, and the deletion of *bcbot1* in three standard strains of *B. cinerea*, indicating that the effect on virulence is strain-dependent [15]. It has also been proven that genes *BcBOT3* and *BcBOT4*, which encode two cytochrome P450 mono-oxygenases, can catalyze regio- and stereo-specific hydroxylations at carbons C-10 and C-4, respectively [16].

In terms of the genes encoding, the number of cytochrome P450 differs between two independent studies (i.e., 57 [3] vs. 61 [12]). Further, most of the sequences in these independent studies do not match. Among of them, a gene encoding NADPH CPR was detected as two different sequences of EQL01332.1 [3] or OSIN4330.1 [12], respectively, which were consistent with those of *Purpureocillium lilacinum* 36-1 [17] by BLAST at GenBank. In this study, the two genes from the two *O. sinensis* genome are named NADPH CPR-1 and NADPH CPR-2, respectively.

The objectives of this study are to (i) analyze the sequence characteristics and structures of the two genes encoding NADPH CPR in *O. sinensis*; (ii) reveal the haplotype diversity and network of the two fungal genes and its host insect COI gene for 219 individuals from 47 *O. sinensis* populations; (iii) conduct a bioinformatic analysis of the phosphorylation sites, motifs, and domains of NADPH CPR encoding by different haplotypes; and (iv) estimate the corresponding relationship between *O. sinensis* and its host insect from the same individual, and discuss the possible influence of diverse gene haplotypes encoding NADPH CPR on the infectivity of *O. sinensis*. Our findings provide solid information to understand the mode of infection of *O. sinensis* into the host insects, which will be useful for the artificial cultivation, protection, and sustainable use of the *O. sinensis*-host insect system.

## 2. Materials and Methods

### 2.1. Sampling

A total of 219 individuals from 47 *O. sinensis* populations were collected, including 25 from Tibet, 17 from Qinghai, 2 from Sichuan, 1 from Yunnan, and 1 from Gansu province, which covered the entire distribution range in Western China (Figure 1 and Figure 2; Appendix A). The collected specimens were identified by Mr. Haifeng Xu from Qinghai Academy of Science and Veterinary Medicine, Qinghai province, China, and then preserved at the Institute of Bioresources and Applied Technology, Tongji University, Shanghai, China. The codes, locations, longitudes, latitudes, altitudes, and sample numbers are shown in Appendix A.

### 2.2. DNA Extraction and Sequencing

Total DNA was extracted independently from the fruiting and caterpillar bodies of each individual using a modified CTAB method [18]. The primers and protocols to amplify the two genes encoding the NADPH CPR of *O. sinensis* are described in Appendix A. The primers were designed and screened based on the NADPH CPR sequences from the two *O. sinensis* genomes of EQL01332.1 [3] and OSIN4330.1 [12]. PCR amplification was carried out with an initial denaturation of 94 °C for 5 min, 35 cycles of 94 °C for 1 min, 57 °C for 30 s, and 72 °C for 1 min, with a final 10 min elongation at 72 °C. Purification and sequencing were conducted by Shanghai Sunny Biotechnology Co. Ltd., Shanghai, China. The sequence GenBank accessions of the two genes encoding NADPH CPR of *O. sinensis* are shown in Appendix A. PCR amplification for the mitochondrial COI gene sequences of the host insect of each individual was performed using our previously described primers and protocols [19]. The detected COI sequences of host insects were compared with GenBank; their accessions are shown in Appendix A.

### 2.3. Data Analyses

The NADPH CPR gene sequences of *O. sinensis* and host insect COI sequences were aligned using MEGA version 7 [20]. Molecular diversity indices, including the number of haplotypes and haplotype frequencies, were calculated using DnaSP version 6.11.01 for each population [21]. Genealogical relationships among the detected haplotypes in each defined group were examined using Network version 10 [22]. A closely related NADPH CPR gene of *Purpureocillium lilacinum* 36-1 (GenBank accession No. LCWV00000000.1) was used as an outgroup [17]. The COI gene sequence of *Papilio torquatus tolmides* Godman and Salvin 1890 (GenBank accession No. JQ606303.1) was selected as an outgroup for the host insects. Genetic distances among populations were determined based on Kimura’s two-parameter model using MEGA version 7. The geographical distances of the *O. sinensis* populations were estimated based on their latitudes and longitudes using the GenAlEx 6.5 software [23]. The correlations between the geographical and genetic distances of the populations were examined using Mantel tests [24]. All Mantel tests were conducted using the GenAlEx 6.5 software with 9999 randomizations (http://biology-assets.anu.edu.au/GenAlEx/Welcome.html) (The Australian National University, Canberra ACT, Australia). The Gene Structure Display Server (GSDS) was used to analyze the differences in gene structure in different populations (http://gsds.cbi.pku.edu.cn). The phosphorylation sites of serine, threonine, or tyrosine were predicted by the NetPhos-3.1 Server (https://services.healthtech.dtu.dk/service.php?NetPhos-3.1). The functional sites of proteins were predicted by PredictProtein (https://www.predictprotein.org/). The Pfam database was used to predict the domains of proteins (http://pfam.xfam.org/). 

## 3. Results

### 3.1. Gene Sequence Characteristics and Protein Structures

Both the NADPH CPR-1 gene and the NADPH CPR-2 gene of *O. sinensis* were detected in 219 individuals of 47 populations. The gene sequences were 2192 and 3603 bp, encoding 692 and 1065 aa, respectively. Their gene structures were characterized by three and seven exons, respectively (see Figure 3). The NADPH CPR-1 gene had an average nucleotide variation rate of 2.05%, containing 39 parsimony informative sites and six singleton sites. The NADPH CPR-2 gene had an average nucleotide variation rate of 1.61%, and 54 parsimony informative sites and four singleton sites were detected in the 3603 bp sequence.

### 3.2. Haplotypes Diversity of NADPH CPR Genes

A total of 28 distinct haplotypes of the NADPH CPR-1 gene of *O. sinensis* were identified from the 219 individuals of 47 populations. Among them, four haplotypes, H1 (76.50%), H13 (4.11%), H3 (2.28%), and H23 (2.28%), with higher frequencies were shared by multiple populations. These four haplotypes were detected in 187 individuals of 45 *O. sinensis* populations, accounting for 85.39% of all individuals and 95.74% of populations (see Figure 1). The haplotype H1 had the broadest distribution across the entire distribution range of *O. sinensis* in Western China, appearing in 76.50% of all individuals and 42 populations (89.36%). The haplotype H13 was mainly distributed in the populations of Tibet (Pop 7, Pop 8 and Pop19), Yunnan (Pop 44) and Sichuan (Pop 46). The haplotype H3 was a unique haplotype distributed in the populations Pop 1, Pop 2, Pop 3 and Pop 4 at the southern margins of the Qinghai Tibet Plateau. The haplotype H23 was only distributed in the population 21 at the Nyingchi area of Tibet. Phylogenetic network analysis of the NADPH CPR-1 gene haplotypes showed that most haplotypes were connected in a star-like manner with the central haplotype H1 (see Figure 4). These haplotypes were considered as dominant haplotypes with a broad distribution. Only six private haplotypes, H5, H8, H14, H16, H17, and H28, were distant from the dominant haplotypes. Only about 5.43% of 129 *O. sinensis* individuals, which were from populations Pop 3, Pop 9, Pop 21 and Pop 24, had the specific private haplotypes of the NADPH CPR-1 gene.

A total of 29 distinct haplotypes of the NADPH CPR-2 gene of *O. sinensis* were identified. Among them, six haplotypes, H1 (50.23%), H17 (9.59%), H11 (7.76%), H19 (5.94%), H22 (4.57%), and H7 (4.11%), with higher frequencies were shared by multiple populations. These six haplotypes of *O. sinensis* were detected in 180 individuals of 43 populations, accounting for 82.19% and 91.49% of all individuals and populations, respectively, with a wide range of distribution (see Figure 2). The most widespread haplotype, H1, with the highest frequency appeared in 74.47% of populations. The haplotype H11 was mainly distributed in the populations except Yunan, i.e., populations Pop 9, Pop 10, Pop 11, Pop 21, Pop 22 and Pop 24 of Tibet, populations Pop 33, Pop 34, Pop 35, Pop38, Pop 39, Pop 41 of Qinghai, and populations Pop 45 and Pop 46 of Sichuan. The haplotype H17 and H19 were mainly distributed in the populations at the areas around The Qinghai Lake (Pop 28, Pop 30, Pop 31, Pop 32) and Middle of Qinghai (Pop 33, Pop 34, Pop 36, and Pop 39). The haplotype H22 was distributed in the populations of Tibet (Pop 20, Pop 21 and Pop 22), Qinghai (Pop 34, Pop 35 and Pop 36), and Sichuan (Pop 45). The haplotype H7 was distributed in the populations Pop 5, Pop 8, Pop 19 and Pop 36 at the Nyingchi area of Tibet. In the phylogenetic network of NADPH CPR-2 gene haplotypes (see Figure 5), these dominant haplotypes with higher frequencies were connected in a star-like manner and haplotype H1 was considered as the central haplotype. Four private haplotypes, H4, H10, H12, and H29, were distant from the dominant haplotypes. Only about 5.43% of 129 *O. sinensis* individuals, which were also from populations Pop 3, Pop 9 and Pop 24, possessed the specific private haplotypes of NADPH CPR-2 gene. Mantel tests showed that the genetic distances of *O. sinensis* populations, based on the two NADPH CPR genes, were not significantly correlated with their geographical distances (r_1_ = −0.025, *p* = 0.330; r_2_ = 0.003, *p* = 0.500).

A total of 59 distinct haplotypes of the host insect COI gene of *O. sinensis* were also identified from the 219 individuals of 47 populations. Among of them, five haplotypes, H26 (21.30%), H15 (15.74%), H81 (12.96%), H35 (6.94%), and H1 (6.02%), were shared by multiple populations. These five haplotypes were detected in 62.96% of all individuals and in 72.34% of all populations. The haplotype H26 had the highest frequency and appeared in 27.66% of all populations. The phylogenetic network indicates that most haplotypes were close to the dominant haplotypes of H1, H15, H26, and H35 (see Figure 6). The haplotype H81 and other private haplotypes were distant from the dominant haplotypes, according to the *Papilio torquatus tolmides* outgroup rooting.

### 3.3. Functional Analysis of NADPH CPR

The phosphorylation sites of all NADPH CPR-1 gene haplotypes coding proteins were evenly distributed throughout the whole peptide chain, mainly serine and threonine phosphorylation sites. There were 24 serine phosphorylation sites (Ser), 25 threonine phosphorylation sites (Thr), and 11 tyrosine phosphorylation sites (Tyr) in the haplotype H1 encoding NADPH CPR-1. However, the phosphorylation sites differed from those coded by the private haplotypes H5, H16, H17, and H28 in NADPH CPR-1; mainly due to the lack of phosphorylation sites at 397 and 401 amino acids, and the addition of phosphorylation sites at 425, 443, and 454 amino acids (Table 1).

All NADPH CPR-2 gene haplotypes coding proteins of *O. sinensis* were mainly phosphorylated by serine and were distributed in the posterior segment of the peptide chain. There were 49 serine phosphorylation sites (Ser), 35 threonine phosphorylation sites (Thr), and 13 tyrosine phosphorylation sites (Tyr) in the haplotype H1 encoding NADPH CPR-2. However, the private haplotypes H4, H10, H12, and H29 lacked phosphorylation sites at 23, 24, 25, and 252 amino acids, and increased the number of phosphorylation sites at 189 and 257 amino acids (Table 2).

The prediction of protein functional sites showed that those NADPH CPR-1 encoding by all haplotypes had three N-glycosylation sites, two cAMP- and cGMP-dependent protein kinase phosphorylation sites, nine Protein kinase C phosphorylation sites, seven Casein kinase II phosphorylation sites, one Tyrosine kinase phosphorylation site, eight N-myristoylation sites, and two Tyrosine kinase phosphorylation sites (Table 3). However, there were differences in the positions, numbers, and amino acid sequences of these sites among NADPH CPR-1 encoding by different haplotypes. Compared with NADPH CPR-1 encoding by dominant haplotypes, all those encoding by private haplotypes (except for haplotype H14) increased one motif, that is, the Casein kinase II phosphorylation site (amino acid positions 425–428) decreased two motifs: the Protein kinase C phosphorylation site (amino acid positions 401–403) and the Casein kinase II phosphorylation site (amino acid positions 401–404). Moreover, the sequence of Tyrosine kinase phosphorylation site 2 (amino acid positions 398–405) changed from RLGSDKDY (in dominant haplotypes) to RLGNDKDY (in private haplotypes).

The NADPH CPR-2 encoding by all haplotypes possessed three N-glycosylation sites, four cAMP- and cGMP-dependent protein kinase phosphorylation sites, thirteen Protein kinase C phosphorylation sites, fourteen Casein kinase II phosphorylation sites, and thirteen N-myristoylation sites, as well as four unique Amidation sites and one unique Cytochrome P450 cysteine heme–iron ligand signature. Similarly, there were certain differences in the positions, numbers, and amino acid sequences of these sites among NADPH CPR-2 encoding by different haplotypes (Table 4). Compared with the NADPH CPR-2 encoding by dominant haplotypes, all those encoding by the private haplotypes increased a motif (i.e., the Protein kinase C phosphorylation site at amino acid positions 189–191) and decreased a motif (i.e., the N-myristoylation site at amino acid positions 461–466). Moreover, all private haplotypes encoding NADPH CPR-2 possessed a unique Cytochrome P450 cysteine heme–iron ligand signature (amino acid positions 400–409), with the peptide amino acid residue of FGNGKRACIG.

The NADPH CPR-1 of *O. sinensis* had three domains, with a typical CPR domain (Figure 7). There was a flavin redox protein domain with FMN binding site (amino acid positions 67–212), an FAD binding site (amino acid positions 268–488), and an NAD binding domain (amino acid positions 544–656). The 117th amino acid of haplotype H1 encoding NADPH CPR-1 at the FMN binding site was alanine (Ala), while the amino acid of all private haplotypes encoding NADPH CPR-1 at this site was valine (Val).

NADPH CPR-2 had four conserved domains (Figure 7). There was a cytochrome P450 domain (amino acid positions 7–456) which was absent in NADPH CPR-1, a flavin redox protein domain with an FMN binding site (amino acid positions 502–636), an FAD binding site (amino acid positions 673–879), and an NAD binding domain (amino acid positions 912–1024). The 539th and 625th amino acids of haplotype H1 encoding NADPH CPR-2 were cysteine (Cys) and glycine (Gly), respectively, while those of all private haplotypes encoding NADPH CPR-2 were tyrosine (Tyr) and aspartic acid (Asp), respectively.

### 3.4. Correspondence between O. sinensis and Its Host Insects

According to the one-to-one correspondence match between *O. sinensis* and its host insect from the same individual, the dominant haplotype H1 and close haplotypes of the NADPH CPR-1 gene or NADPH CPR-2 gene could infect almost all host insects, while the private haplotypes H5, H6, H17, and H76 could infect few host insects (Appendix A). Those host insects corresponding to the private haplotypes could also be infected by the dominant haplotypes of NADPH CPR genes of *O. sinensis*.

## 4. Discussion

Whole-genome sequencing can reveal the genetic information of species, clarify the regulation mode of gene expression, understand the molecular evolution process of species, and explore the molecular mechanisms of species adapting to the environment [25]. It has been widely used in human beings [26], rice [27], etc. However, two independent studies previously carried out on the whole genome sequencing of *O. sinensis* are not consistent in terms of fungal pathogenicity. They even had opposing results in the expansion or contraction of the functional gene family related to fungal pathogenicity and, consequently, the infection mechanism of *O. sinensis* [3,12]. This may have been due to the scaffold N50 value, actual assembled genome size, and coverage of the estimated genome size by different genome sequencing technologies [3,12]. In this study, two genes encoding NADPH CPR were detected in all 219 individuals from 47 *O. sinensis* populations, while only one gene encoding NADPH CPR had been detected in the above two independent studies [3,12]. Understanding how to find biologically significant information from the massive whole genome data has become both very difficult and very important.

The average nucleotide variation rates of the two NADPH CPR gene sequences of *O. sinensis* were 2.05% and 1.61%, respectively, much lower than the rate of 8.4% of the fungal 6-locus sequences of nrDNA ITS, *MAT1-2-1*, *csp1*, OSRC14, OSRC17, and OSRC27 [28]. These conserved gene sequences may be related to the protein functions of NADPH CPR. The sequence homologies of two NADPH CPR genes of *O. sinensis*, corresponding to those of *Purpureocillium lilacinum* 36-1, were 82.45% and 72.58%, respectively. Their gene structures were identical, although the gene sequence length was different [17].

The genetic differences among most haplotypes of the two NADPH CPR genes of *O. sinensis* were small, with the dominant haplotypes (such as H1 and close haplotypes) being widely distributed throughout its entire distribution range in Western China. This was also supported by the result that there was no significant correlation between genetic distances based on the two genes and the geographical distances of *O. sinensis* populations. Only a few specific private haplotypes of the two genes were distributed in the limited *O. sinensis* populations Pop 3, Pop 9, Pop 21 and Pop 24. Those populations are located in the Nyingchi region and at the boundary of the Qinghai Tibet Plateau, which are usually considered to be the origin and diversity centers of *O. sinensis* [19,28,29].

Bioinformatic analysis predicted that the structure and function of NADPH CPR of *O. sinensis* were different between those encoding by the dominant haplotypes and private haplotypes. There were differences in the phosphorylation sites and quantities of NADPH CPR of *O. sinensis* between those encoding the dominant haplotypes and private haplotypes. Such changes in the phosphorylation sites may affect the binding of proteins and cofactors, and thus change the protein activity. The replacement of Tyr-178 with aspartate abolished FMN binding. The FAD level in the Asp-178 mutant was also decreased, suggesting that FAD binding is dependent upon FMN [30].

The motif numbers and amino acids sequences of NADPH CPR of *O. sinensis* differed between those encoding by the dominant haplotypes and private haplotypes. The increase or decrease in these motifs and amino acid sequence differences may change the protein function and affect the pathogenicity of the fungi. Even subtle changes in the composition of a motif can result in significant changes in affinity, which can be further modulated by phosphorylation [31]. In this study, the NADPH CPR-2 encoding by the private haplotypes of *O. sinensis* possessed a unique cysteine heme–iron ligand signature (amino acid positions 400–409) with an amino acid residue of FGNGKRACIG, which is common to all P450s and is essential for catalytic activity [32].

The two NADPH CPR of *O. sinensis* have typical domains, consisting of a flavoprotein containing binding domains for FMN, FAD, and NADPH [33]. As part of the redox reaction, an NADPH CPR is needed for transferring electrons to P450s [34]. In this study, there were amino acid substitutions in the motifs of the FMN domain in the two NADPH CPR of *O. sinensis* for those encoded by the dominant haplotypes and private haplotypes. The FMN domain mutations might improve the interaction with a specific CYP using several binding motifs of the FMN domain [14]. Therefore, there may be functional changes between the NADPH CPR of *O. sinensis* encoding by dominant and private gene haplotypes.

There were three corresponding patterns between *O. sinensis* and its host insect of the same individual: (i) the dominant NADPH CPR gene haplotypes of *O. sinensis* vs. the dominant COI gene haplotypes of its host insect; (ii) the dominant fungal NADPH CPR gene haplotypes vs. the private host insect COI gene haplotypes; and (iii) the private fungal NADPH CPR gene haplotypes vs. the private host insect COI gene haplotypes. This means that *O. sinensis* with the dominant NADPH CPR gene haplotypes could infect almost all host insects. Another study has also suggested that one fungal multiple locus haplotypes (MLH) was capable of infecting multiple host insect MLHs [28]. The fungus *O. sinensis* has been confirmed as a single species based on the rDNA-ITS sequence [19,35,36,37], while 57 species representing seven genera (*Hepialus*, *Thitarodes*, *Pharmacis*, *Magnificus*, *Gazoryctra*, *Endoclita*, and *Bipectilus*) are possible hosts of *O. sinensis* [38]. The ascospores of *O. sinensis* can disperse over a large area, through air, rain, and host insects [39]. The host insects are poorly adapted for long-distance flight and dispersal, and the adults survive for only 3–8 days [40]. The intraspecific genetic differentiation of *O. sinensis* and interspecific genetic differentiation of its host insects, as well as the difference in spread characteristics between the fungus and its host insects, suggests that the widely distributed *O. sinensis* with the NADPH CPR gene H1 and close haplotypes may strongly infect almost all host insects through random infection by oral or respiratory pores. A genome survey uncovered that some genes related to cuticle degrading proteases and adhesion to cuticles had been greatly reduced in size, suggesting that *O. sinensis* can no longer breach the intact cuticle and, instead, probably infects insects orally or via the spiracles [3]. Other studies have also supported the idea that *O. sinensis* infects host insects through oral activity [6].

In this study, the *O. sinensis* with private haplotypes of the NADPH CPR gene could only infect very few specific host insects. It is speculated that the specific infection might be through breaching the cuticle of host insects with weak infectivity, as specificity is usually controlled by infection events at the level of the cuticle [41]. Similar to biotrophic plant pathogens [42], *O. sinensis* has a genome shaped by retrotransposon-driven expansions, suggesting the biphasic pathogenic mechanism of fungus beginning with stealth pathogenesis in early host instars and a lethal stage in late instars [3]. The infection rate of *O. sinensis* in host insects was only 14.06% by cuticle coating inoculation, much lower than those by cuticle injection, oral injection, or mixed with feed [10]. In nature, the cuticle infection of *O. sinensis* in its host insects may not be the major route.

Cytochrome P450s play essential roles in fungal physiologies, including detoxification, degradation of xenobiotics, and the biosynthesis of secondary metabolites [43]. The less specific infections of *O. sinensis* with the private haplotypes was revealed based on the NADPH CPR, which is related to fungal pathogenicity [17]. The difference in phosphorylation sites, motifs, and domains of NADPH CPR of *O. sinensis* may lead to functional changes in those encoded by the private haplotypes. The infection modes and mechanisms of *O. sinensis* in its host insects should be explored through more genome, transcriptome, and protein analyses.

## Figures and Tables

**Figure 1 microorganisms-08-00968-f001:**
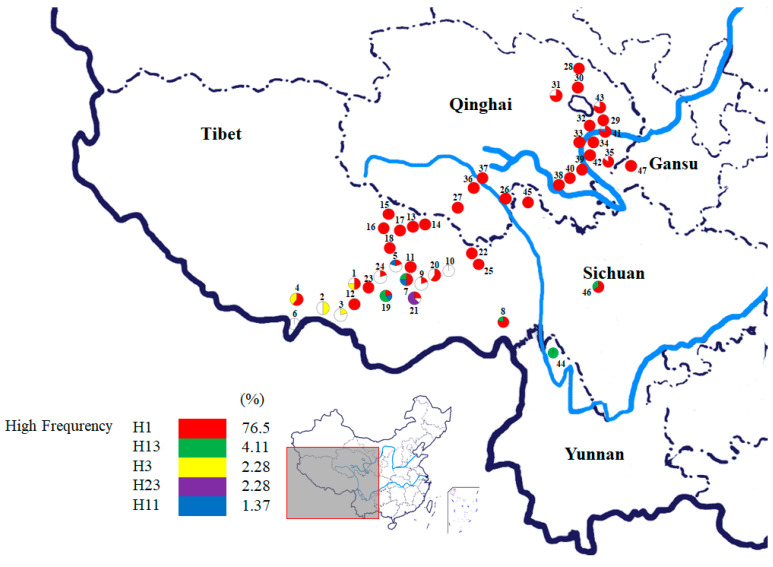
Geographical distribution map and NADPH-cytochrome P450 reductase-1 gene haplotype frequencies of 47 *O. sinensis* populations collected across the major distribution regions in China. Population codes correspond to those in Appendix A. The haplotypes with high frequency are color-coded, while haplotypes with low frequency are represented by white circles.

**Figure 2 microorganisms-08-00968-f002:**
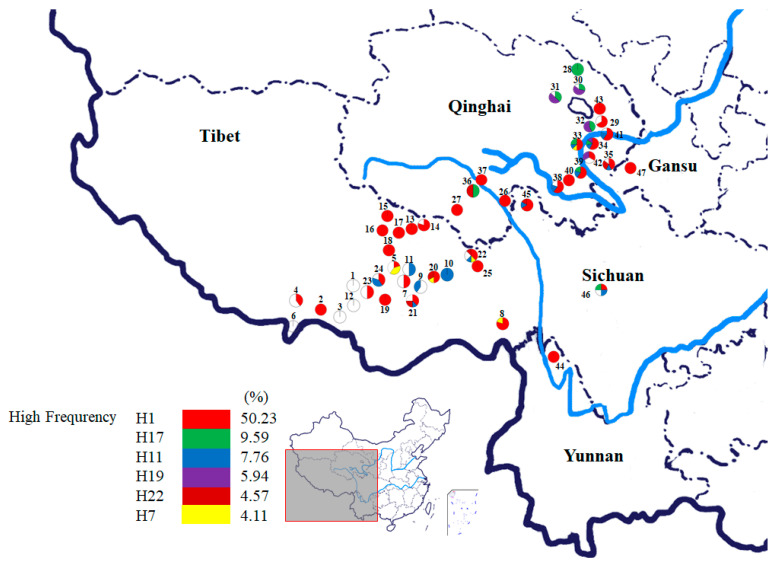
Geographical distribution map and NADPH-cytochrome P450 reductase-2 gene haplotype frequencies of 47 *O. sinensis* populations collected across the major distribution regions in China. Population codes correspond to those in Appendix A. The haplotypes with high frequency are color-coded, while haplotypes with low frequency are represented by white circles.

**Figure 3 microorganisms-08-00968-f003:**
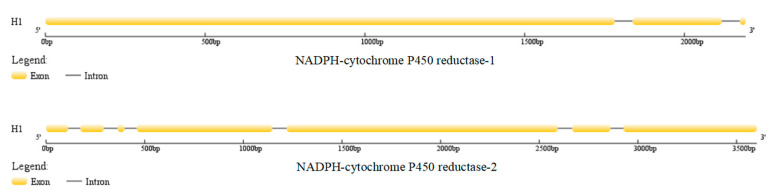
Structure of NADPH-cytochrome P450 reductase of *O. sinensis*.

**Figure 4 microorganisms-08-00968-f004:**
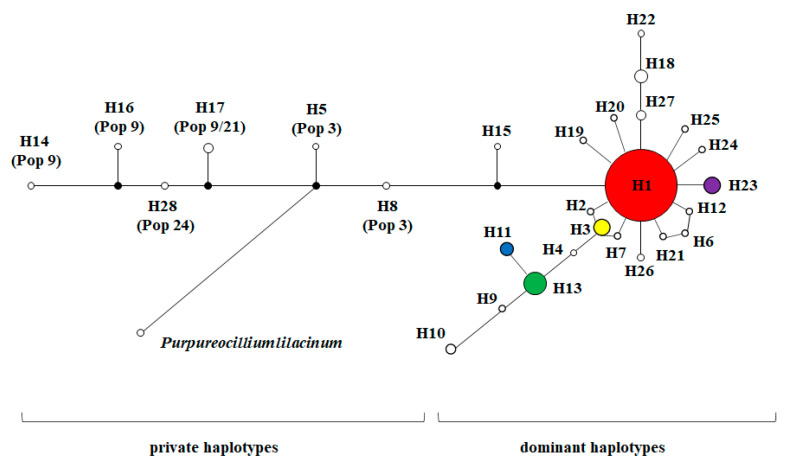
Phylogenetic network of NADPH-cytochrome P450 reductase-1 gene haplotypes of *O. sinensis*. Circle areas are proportional to haplotype frequencies. Circle colors are proportional to haplotypes defined in Figure 1. *Purpureocillium lilacinum* 36-1 (GenBank accession No. LCWV00000000.1) was used as an outgroup. Population codes correspond to those in Appendix A.

**Figure 5 microorganisms-08-00968-f005:**
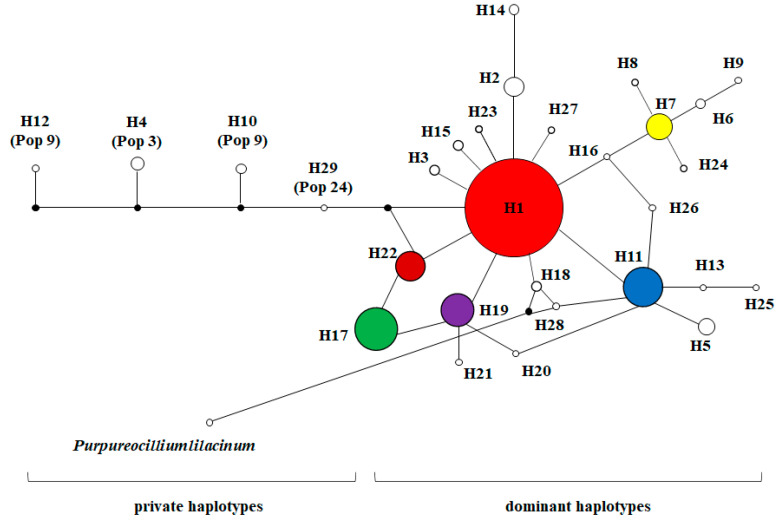
Phylogenetic network of NADPH-cytochrome P450 reductase-2 gene haplotypes of *O. sinensis*. Circle areas are proportional to haplotype frequencies. Circle colors are proportional to haplotypes defined in Figure 2. *Purpureocillium lilacinum* 36-1 (GenBank accession No. LCWV00000000.1) was used as an outgroup. Population codes correspond to those in Appendix A.

**Figure 6 microorganisms-08-00968-f006:**
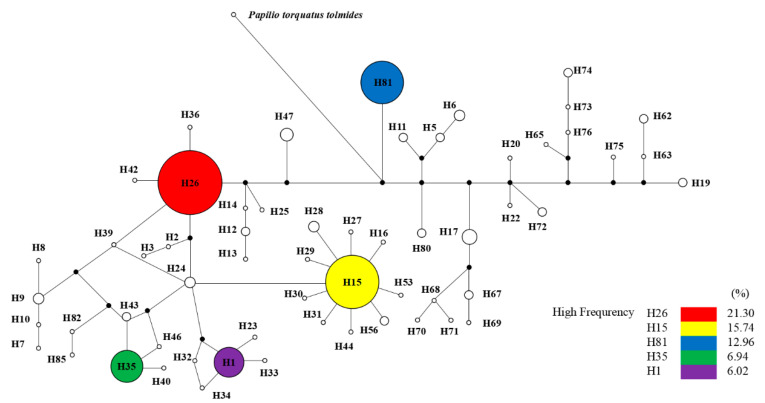
Phylogenetic network of COI gene haplotypes of host insects of *O. sinensis*. Circle areas are proportional to haplotype frequencies. *Papilio torquatus tolmides* (GenBank accession No. JQ606303.1) was an outgroup.

**Figure 7 microorganisms-08-00968-f007:**
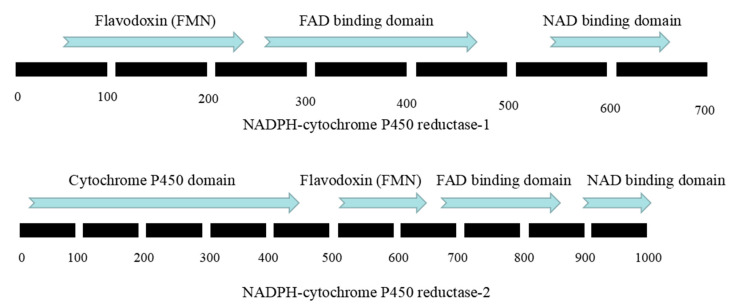
Amino acid sequence domains of NADPH-cytochrome P450 reductases of *O. sinensis*.

**Table 1 microorganisms-08-00968-t001:** Phosphorylation sites of NADPH-cytochrome P450 reductase-1 encoding by different haplotypes of *O. sinensis*.

Position	H1	H5	H8	H14	H16	H17	H28
335	T	T	-	T	-	T	T
397	T	-	-	T	-	-	-
401	S	-	-	S	-	-	-
425	-	S	S	-	S	S	S
443	-	T	T	-	T	T	T
454	-	S	S	-	S	S	S
636	T	T	T	-	-	T	-

S = Ser; T = Thr; - No phosphorylation site at this location.

**Table 2 microorganisms-08-00968-t002:** Phosphorylation sites of NADPH-cytochrome P450 reductase-2. encoding by different haplotypes of *O. sinensis*.

Position	H1	H4	H10	H12	H29
23	T	-	-	-	-
24	T	-	-	-	-
25	S	-	-	-	-
189	-	S	S	S	S
252	S	-	-	-	-
257	-	T	T	T	T
437	-	Y	Y	Y	-
539	-	Y	Y	Y	-
777	S	-	S	S	S
881	S	S	-	-	S

S = Ser; T = Thr; Y = Tyr; - No phosphorylation site at this location.

**Table 3 microorganisms-08-00968-t003:** Predicted protein motifs of NADP. H-cytochrome P450 reductase-1 encoding by different haplotypes of *O. sinensis*.

Functional Site	H1		H5		H8		H14		H16		H17/H28	
	Position	Sequence	Position	Sequence	Position	Sequence	Position	Sequence	Position	Sequence	Position	Sequence
N-gly	167–170	NNTY	167–170	NNTY	167–170	NNTY	167–170	NNTY	167–170	NNTY	167–170	NNTY
	244–247	NLTR	244–247	NLTR	244–247	NLTR	244–247	NLTR	244–247	NLTR	244–247	NLTR
	305–308	NLTY	305–308	NLTY	305–308	NLTY	305–308	NLTY	305–308	NLTY	305–308	NLTY
cA/GMP	461–464	KKIS	461–464	KKIS	461–464	KKIS	461–464	KKIS	461–464	KKIS	461–464	KKIS
	581–584	RKRT	581–584	RKRT	581–584	RKRT	581–584	RKRT	581–584	RKRT	581–584	RKRT
PK C	61–63	SGK	61–63	SGK	61–63	SGK	61–63	SGK	61–63	SGK	61–63	SGK
	113–115	SDK	113–115	SDK	113–115	SDK	113–115	SDK	113–115	SDK	113–115	SDK
	265–267	TAK	265–267	TAK	265–267	TAK	265–267	TAK	265–267	TAK	265–267	TAK
	335–337	TAK	335–337	TAK	335–337	TAK	335–337	TAK			335–337	TAK
	343–345	SVK	343–345	SVK	343–345	SVK	343–345	SVK	343–345	SVK	343–345	SVK
	350–352	TAK	350–352	TAK	350–352	TAK	350–352	TAK	350–352	TAK	350–352	TAK
	401–403	SDK					401–403	SDK				
	615–617	SKK	615–617	SKK	615–617	SKK	615–617	SKK	615–617	SKK	615–617	SKK
	636–638	TQK			636–638	TQK					636–638	TQK
CK II	75–78	TAED	75–78	TAED	75–78	TAED	75–78	TAED	75–78	TAED	75–78	TAED
	123–126	TYGE	123–126	TYGE	123–126	TYGE	123–126	TYGE	123–126	TYGE	123–126	TYGE
	141–144	TADD	141–144	TADD	141–144	TADD	141–144	TADD	141–144	TADD	141–144	TADD
	204–207	TMEE	204–207	TMEE	204–207	TMEE	204–207	TMEE	204–207	TMEE	204–207	TMEE
	359–362	TTFD	359–362	TTFD	359–362	TTFD	359–362	TTFD	359–362	TTFD	359–362	TTFD
	401–404	SDKD					401–404	SDKD				
	591–594	SEWE	591–594	SEWE	591–594	SEWE	591–594	SEWE	591–594	SEWE	591–594	SEWE
			425–428	SKGE	425–428	SKGE			425–428	SKGE	425–428	SKGE
TK 1	582–589	KRTEDFLY	582–589	KRTEDFLY	582–589	KRTEDFLY	582–589	KRTEDFLY	582–589	KRTEDFLY	582–589	KRTEDFLY
N-myr	26–31	GTYWGV	26–31	GTYWGV	26–31	GTYWGV	26–31	GTYWGV	26–31	GTYWGV	26–31	GTYWGV
	45–50	GVKAGR	45–50	GVKAGR	45–50	GVKAGR	45–50	GVKAGR	45–50	GVKAGR	45–50	GVKAGR
	70–75	GSQTGT	70–75	GSQTGT	70–75	GSQTGT	70–75	GSQTGT	70–75	GSQTGT	70–75	GSQTGT
	150–155	GNDPAL	150–155	GNDPAL	150–155	GNDPAL	150–155	GNDPAL	150–155	GNDPAL	150–155	GNDPAL
	164–169	GLGNNT	164–169	GLGNNT	164–169	GLGNNT	164–169	GLGNNT	164–169	GLGNNT	164–169	GLGNNT
	303–308	GSNLTY	303–308	GSNLTY	303–308	GSNLTY	303–308	GSNLTY	303–308	GSNLTY	303–308	GSNLTY
	483–488	GVATNY	483–488	GVATNY	483–488	GVATNY	483–488	GVATNY	483–488	GVATNY	483–488	GVATNY
	568–573	GLDVGR	568–573	GLDVGR	568–573	GLDVGR	568–573	GLDVGR	568–573	GLDVGR	568–573	GLDVGR
TK 2	398–405	RLGSDKDY	398–405	RLGNDKDY	398–405	RLGNDKDY	398–405	RLGSDKDY	398–405	RLGNDKDY	398–405	RLGNDKDY
	582–589	KRTEDFLY	582–589	KRTEDFLY	582–589	KRTEDFLY	582–589	KRTEDFLY	582–589	KRTEDFLY	582–589	KRTEDFLY

N-gly = N-glycosylation site; cA/GMP = cAMP- and cGMP-dependent protein kinase phosphorylation site; PK C = Protein kinase C phosphorylation site; CK II = Casein kinase II phosphorylation site; TK 1 = Tyrosine kinase phosphorylation site 1; N-myr = N-myristoylation site; TK 2 = Tyrosine kinase, phosphorylation site 2; Bold indicates difference compared with H1.

**Table 4 microorganisms-08-00968-t004:** Predicted protein motifs of NADPH-cytochrome P450 reductase-2 encoding by different haplotypes of *O. sinensis*.

Functional Site	H1		H4		H10/H12		H29	
	Position	Sequence	Position	Sequence	Position	Sequence	Position	Sequence
N-gly	255–258	NITD	255–258	NITD	255–258	NITD	255–258	NITD
	429–432	NFSL	429–432	NFSL	429–432	NFSL	429–432	NFSL
	734–737	NLSW	734–737	NLSW	734–737	NLSW	734–737	NLSW
cA/GMP	190–193	KRPS	190–193	KRPS	190–193	KRPS	190–193	KRPS
	225–228	RKES	225–228	RKES	225–228	RKES	225–228	RKES
	670–673	RKVT	670–673	RKVT	670–673	RKVT	670–673	RKVT
	809–812	KRTS	809–812	KRTS	809–812	KRTS	809–812	KRTS
PK C	112–114	SIR	112–114	SIR	112–114	SIR	112–114	SIR
	230–232	SGR	230–232	SGR	230–232	SGR	230–232	SGR
	323–325	TLR	323–325	TLR	323–325	TLR	323–325	TLR
	446–448	TIK	446–448	TIK	446–448	TIK	446–448	TIK
	457–459	SLR	457–459	SLR	457–459	SLR	457–459	SLR
	673–675	TLR	673–675	TLR	673–675	TLR	673–675	TLR
	685–687	SAR	685–687	SAR	685–687	SAR	685–687	SAR
	694–696	SVK	694–696	SVK	694–696	SVK	694–696	SVK
	708–710	TYR	708–710	TYR	708–710	TYR	708–710	TYR
	740–742	TLK	740–742	TLK	740–742	TLK	740–742	TLK
	745–747	SDR	745–747	SDR	745–747	SDR	745–747	SDR
	774–776	TKR	774–776	TKR	774–776	TKR	774–776	TKR
	822–824	SIK	822–824	SIK	822–824	SIK	822–824	SIK
			189–191	SKR	189–191	SKR	189–191	SKR
CK II	54–57	STNE	54–57	STNE	54–57	STNE	54–57	STNE
	89–92	TAYD	89–92	TAYD	89–92	TAYD	89–92	TAYD
	246–249	TTGD	246–249	TTGD	246–249	TTGD	246–249	TTGD
	431–434	SLDD	431–434	SLDD	431–434	SLDD	431–434	SLDD
	463–466	TPTE	463–466	TPTE	463–466	TPTE	463–466	TPTE
	476–479	TAAE	476–479	TAAE	476–479	TAAE	476–479	TAAE
	568–471	SWME	568–471	SWME	568–471	SWME	568–471	SWME
	631–634	SAFE	631–634	SAFE	631–634	SAFE	631–634	SAFE
	657–660	SGND	657–660	SGND	657–660	SGND	657–660	SGND
	758–761	SASD	758–761	SASD	758–761	SASD	758–761	SASD
	812–815	SILD	812–815	SILD	812–815	SILD	812–815	SILD
	859–862	SVLE	859–862	SVLE	859–862	SVLE	859–862	SVLE
	881–884	SGLE	881–884	SGLE			881–884	SGLE
	950–953	SPDE	950–953	SPDE	950–953	SPDE	950–953	SPDE
N-myr	82–87	GVHDGL	82–87	GVHDGL	82–87	GVHDGL	82–87	GVHDGL
	242–247	GVDPTT	242–247	GVDPTT	242–247	GVDPTT	242–247	GVDPTT
	461–466	GMTPTE					461–466	GMTPTE
	491–496	GIAHTR	491–496	GIAHTR	491–496	GIAHTR	491–496	GIAHTR
	505–510	GSNSGT	505–510	GSNSGT	505–510	GSNSGT	505–510	GSNSGT
	557–562	GQPPSN	557–562	GQPPSN	557–562	GQPPSN	557–562	GQPPSN
	580–585	GVSYAV	580–585	GVSYAV	580–585	GVSYAV	580–585	GVSYAV
	650–655	GTEDSS	650–655	GTEDSS	650–655	GTEDSS	650–655	GTEDSS
	693–698	GSVKNH	693–698	GSVKNH	693–698	GSVKNH	693–698	GSVKNH
	828–833	GSYLGM	828–833	GSYLGM	828–833	GSYLGM	828–833	GSYLGM
	874–879	GVATSF	874–879	GVATSF	874–879	GVATSF	874–879	GVATSF
	882–887	GLEPGE	882–887	GLEPGE	882–887	GLVPGE	882–887	GLEPGE
	1035–1040	GCKQGW	1035–1040	GCKQGW	1035–1040	GCKQGW	1035–1040	GCKQGW
Ami	185–188	CGKR	185–188	CGKR	185–188	CGKR	185–188	CGKR
	230–233	SGRK	230–233	SGRK	230–233	SGRK	230–233	SGRK
	402–405	NGKR	402–405	NGKR	402–405	NGKR	402–405	NGKR
	935–938	AGRK	935–938	AGRK	935–938	AGRK	935–938	AGRK
CYP450			400–409	FGNGKRACIG	400–409	FGNGKRACIG	400–409	FGNGKRACIG

N-gly = N-glycosylation site; cA/GMP = cAMP- and cGMP-dependent protein kinase phosphorylation site; PK C = Protein kinase C phosphorylation site; CK II = Casein kinase II phosphorylation site; N-myr = N-myristoylation site; Ami = Amidation site; CYP450 = Cytochrome P450 cysteine heme-iron ligand signature; Bold indicates difference compared with H1.

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
