# Peer review of "Haplotype Diversity of NADPH-Cytochrome P450 Reductase Gene of Ophiocordyceps sinensis and the Effect on Fungal Infection in Host Insects"

_microorganisms, 2020, doi:10.3390/microorganisms8070968_

Round 1

Reviewer 1 Report

This is the second time of review for this manuscript, but I don’t think the manuscript has been improved notably.

  1. The authors responded that they performed the basic population genetic analysis. However, there are none of the results of the summary of the analysis. Also, fixation indices were not analyzed.

  1. Annotation on the network (Figure 4-5) is not enough to understand the population structure of each gene. But, the authors did not improve it. Figure 1-2 are not enough to be used. I suggest the software PopART or Network for annotation the population (or region) information on each haplotype (circle).

L153-154, 170-171: My comment is not how to set the outgroup, but setting outgroup did not say anything about the distance from the dominant haplotype. Distance is calculated solely based on the sequence difference between the haplotype pair. Outgroup rooting did not influence on the distance of the haplotype pair.

Author Response

1. This is the second time of review for this manuscript, but I don’t think the manuscript has been improved notably.

(#) Thank you very much for your comments and suggestions on our manuscript. We tried to revise the manuscript according to the reviewer’s comments and suggestions.

2. The authors responded that they performed the basic population genetic analysis. However, there are none of the results of the summary of the analysis. Also, fixation indices were not analyzed.

(#) We did perform the basic population genetic analysis, but did not include in the manuscript for two reasons. Firstly, the haplotype diversity of O. sinensis individuals is able to analyze the effect on fungal infection to host insects according to the one-to-one correspondence match between O. sinensis and its host insect from the same individual. On the contrary, the population genetic analysis is focused on the genetic differentiation of populations, which does not support our study. Secondly, compared to our previous studies and other related studies, the results of population genetic analysis cannot provide more valuable information.

3. Annotation on the network (Figure 4-5) is not enough to understand the population structure of each gene. But, the authors did not improve it. Figure 1-2 are not enough to be used. I suggest the software PopART or Network for annotation the population (or region) information on each haplotype (circle).

(#) Following the suggestion, we have added the population information for those dominant haplotypes with higher frequency. (Line 149-153, Line 169-177)

We have added the population information for the specific private haplotypes in Fig.4 and Fig.5.

4. L153-154, 170-171: My comment is not how to set the outgroup, but setting outgroup did not say anything about the distance from the dominant haplotype. Distance is calculated solely based on the sequence difference between the haplotype pair. Outgroup rooting did not influence on the distance of the haplotype pair.

(#) We have deleted the sentence “according to the Purpureocillium lilacinum 36-1 outgroup rooting”. (Line 155, line 180)

We have doublechecked all data, including population information and haplotypes, and also corrected some statements throughout the text. (Line 156-158, line 181, line 317, line 15)

Reviewer 2 Report

All comments are incorporated in the manuscript.

Author Response

Thank you very much for your comments and suggestions on our manuscript.

Reviewer 3 Report

The authors have implemented most of the earlier comments.

Author Response

(The authors gave the same response as above.)

Round 2

Reviewer 1 Report

I agree the author's responses and suggest to publish in this form.

This manuscript is a resubmission of an earlier submission. The following is a list of the peer review reports and author responses from that submission.

Round 1

Reviewer 1 Report

An interesting collection of data has been collected on cytochrome P450 reductase gene sequences of a fungus which apparently is used to combat a root-boring lepidopteran pest. The data show a pattern of haplotype diversity across the country, however the relavanve of all this escapes me.

This paper needs a thorough update on English style because it is difficult to understand and the poor English style makes it only worse.

In addition, the paper lacks any context or problem statement. Although the data seem to be interesting and relevant, the reader is completely left in the dark to decide what is the importance and what problem was solved through this study.

  • The Abstract mentions something about a traditional medicine, but this does not return anywhere in the paper.
  • What was the aim of the study? The objectives (line 66) are formulated in terms of what you have done, but we need objectives as to the rationale of the study.
  • What is the agricultural context in which you are interested in the fungus-insect relationship. I assume the hepialids are a pest, but on what crop?
  • Is the fungus considered as a control agent?
  • Why did you select the P450 reductase gene; is it just a marker, or does it bear any relationship with the virulence or the resistance of the fungus; if so what relationship?
  • Why the analysis of phosphorylation sites is done in this study totally escapes me. It must have something to do with the function of the gene, but what is the context of the analysis and the relevance remains a mystery to me.
  • How did you isolate the DNA sequence of the CPR genes; are CPR-1 and CPR-2 the only paralogs in the genome? Does the sequence cover the complete open reading frame, if not what part of the gene did you analyse?
  • How do you define the haplotypes. Is every nucleotide polymorphism in the DNA sequence a new haplotype?
  • I can’t understand why COI from another species can be used as an outgroup for CPR sequence analysis.

Reviewer 2 Report

This is a good contribution on pathogenesis of Ophiocordyceps sinensis on hepialid larvae. However, some discussion on fungal pathogenesis on plants would add value to the manuscript.

The authors have mostly used past tense in Introduction and the rest also. Natural phenomena are better expressed in present tense.

Italicize scientific names throughout the text, including references.

Please improve the grammar of the manuscript.

Lines 75-77, please mention how the specimens were identified as O. sinensis. Correct identification of biological specimens is always of utmost importance to draw any conclusion.

Line 302, please provide more recent references.

Lines 347-349, what is the year of publication?

Line 355, what is Publisher correction?

Line 449, it should be Mycosystema

Line 452-454, what is the page no.?

Reviewer 3 Report

The manuscript investigated the diversity and predicted function of NADPH-cytochrome P450 reductase (CPR) from fungal pathogen Ophiocordyceps sinensis and host Hepialidae. Ophiocordyceps sinensis is an important source of traditional medicine and the CPR gene has known to be associated with host infection. CPR-1 and CPR-2 gene has consisted of a few dominant haplotypes and the network looks like a star-shaped structure. Phosphorylation sites, motifs, and domains of the CPR gene showed different characteristics between dominant and rare haplotype. Also, the dominant haplotype may have a strong infection. Considering the importance of Ophiocordyceps sinensis, this study has a value on the biology and pharmacology. However, some analyses need to perform to obtain additional information.

  1. The first part of the manuscript focused on the population genetic analysis. However, authors did not perform basic summary or analysis for characteristics of populations. For example, haplotype diversity or neutrality test results can provide essential information. In addition, a comparison of genetic characteristics between the group (e.g. country) can show some insight into the Ophiocordyceps sinensis biology or ecology. Fixation indices also help to check the differentiation of populations.

  1. Network analysis only showed the relationship between haplotypes. However, the additional information (e.g. country or geographic distribution) can help to understand the readers. Of course, Figure 1-2 contains this kind of information but it is hard to understand in terms of the association between haplotype relationship and geographic distribution.

Line number 53: Please provide a full name for FAD and FMN.

LN 60: Please give a space after 61.

LN 108: Please give a space after 10.

LN 110: Please remove the additional “GenBank”.

LN 144, 160: Outgroup did associate with the state “private” or “distant from the dominant haplotypes”. Outgroup did not show anything for this kind of information.

LN 176: H1, H26, and H35 were not a star-like shape.

Supporting information Table S1. Please give space to the location name appropriately.

Supporting information Table S7. Please change CIO to COI.